## EQUITY, DIVERSITY AND INCLUSION

# Making conferences in the plant sciences more inclusive through community recommendations

**Abstract** An unwelcoming climate and culture at scientific conferences is an obstacle to retaining scientists with marginalized identities. Here we describe how a number of professional societies in the plant sciences, mostly based in the United States, collaborated on a project called ROOT & SHOOT (short for Rooting Out Oppression Together and SHaring Our Outcomes Transparently) to make conferences in the field more inclusive. The guidelines we developed, and our efforts to implement them in 2023 and 2024, are summarized here to assist other conference organizers with creating more inclusive conferences.

MARCIA PUIG-LLUCH, MARY WILLIAMS, ERIC WADA, IMEÑA VALDES, CARRIE TRIBBLE, ANDREW READ, KANWARDEEP S RAWALE, CHELSEA L NEWBOLD, BATHABILE MTHOMBENI, MICHAEL MOODY, LAURA MINERO, ANNARITA MARRANO, MELANIE LINK-PEREZ, ROGER W INNES, CODY COYOTEE HOWARD, ADRIANA HERNANDEZ, CORRI HAMILTON, DENITA HADZIABDIC, MORGAN GOSTEL, JOANNA FRIESNER, JOHN E FOWLER, MINDY FINDLATER, SAKINA ELSHIBLI, STEVEN J BURGESS, HANK W BASS, BURCU ALPTEKIN, R SHAWN ABRAHAMS, PATRICIA BALDRICH\*

**\*For correspondence:** pbaldrich@ucdavis.edu

**Competing interest:** The authors declare that no competing interests exist.

## Introduction

Conferences are an integral part of scientific communication, serving as a vital platform for fostering collaboration, networking opportunities, and showcasing innovations and advances in the field (*Leochico et al., 2021*; *Joo et al., 2022*). However, it is crucial to acknowledge that these opportunities are often not equally accessible or inclusive for all members of the scientific community. Issues stemming from white supremacy and cis-hetero-normativity, inaccessible facilities, unwelcoming environments, and biases in speaker invitations are among the factors perpetuating inequities within our professional meeting spaces (*Jack-Scott, 2023*). These inequities manifest in both explicit and implicit conference practices. Although the science, technology, engineering, and mathematics (STEM) workforce in the United States has diversified in the last decade to include an increased representation of women and underrepresented minorities (*NCSES, 2024*), there is still much work to be done towards the goal of creating safe, inclusive working spaces for all.

In February 2021, the National Science Foundation (NSF) released a Dear Colleague Letter: **LEA**ding cultural change through **P**rofessional **S**ocieties (LEAPS) of Biology. Its goal was to solicit proposals from professional societies "to develop collaborative networks for facilitating cultural changes in the biological sciences to advance diversity, equity, and inclusion" (*NSF, 2021*). A group of several plant science professional societies and organizations collaboratively wrote and submitted a proposal for a Research Coordination Network called ROOT & SHOOT (Rooting Out Oppression & Sharing Our Outcomes Transparently), which was funded later in 2021. The societies involved were the American Society of Plant Biologists, the American Society of Plant

Taxonomists, the American Phytopathological Society, the Botanical Society of America, the International Society for Molecular Plant-Microbe Interactions, Maize Genetics Corporation, and the North American *Arabidopsis* Steering Committee (see *Table 1*). The proposal set out an ambitious set of objectives, including a commitment to work towards safer and more inclusive conferences. Hostile cultures persist in research and academic settings, and conferences can be particularly risky places (*Berhe et al., 2022*). Although ROOT & SHOOT acknowledges that its organization's membership may have an international component, the objectives are considered through a USA-centered perspective.

A central feature of the ROOT & SHOOT efforts to improve inclusivity is the formation of working groups to tackle specific issues. This approach aligns with the project goals of valuing and incorporating the perspectives and ideas of plant scientists from communities that are underrepresented and historically excluded from science. Here, we report on the formation of the ROOT & SHOOT Inclusive Conference Working Group, its recommendations, and the results of our initial efforts to implement them. We currently have a second working group underway, focused on culturally responsible mentorship, and are currently developing a seven-week course that is being prepared for beta testing.

## Working groups

The working group approach has several advantages: the scope of the task can be focused and accomplished within a defined time frame; it harnesses the knowledge and energy of community members with relevant expertise and interest; and it incorporates a consensus-trust decision-making approach (*Dong et al., 2021*; *Gai et al., 2023*). The working group process also makes it possible to recruit members who do not belong to traditional power structures, such as early-career scholars who are unlikely to have yet obtained a leadership position in a professional society but may have previously unheard views (*Smith and Turner, 2015*; *Bialik and Fry, 2019*; *Robbins et al., 2022*). In the long term, a working group process fosters a more inclusive future by developing a robust network of diversity, equity, and inclusion advocates across disciplines in the plant sciences, and by enabling participants to gain valuable leadership and consensus-building experience.

### Participant selection

To correct past injustices and better represent the diversity within our organizations, representatives from ROOT & SHOOT deliberately aimed to recruit a diverse group of volunteers from the participating societies by using an open call. Early-career researchers were considered essential contributors to the working group process. Therefore, instead of standard application inputs (e.g., CV, research experience), applicants were asked for responses to free-form questions seeking to understand their motivation and life experiences that could be relevant to the desired outcomes. This application was shared with all the members of the participating societies, the ROOT & SHOOT mailing list, and social media. To help facilitators achieve a representative set of participants, optional questions seeking demographic information were also part of the application. Although it was not a requirement, most (17 of 24) of the Working Group participants had previous conference-organizing experience. These experiences ranged from organizing informal networking events and smaller workshops to professionally planning and hosting several in-person and virtual conferences with

**Table 1.** Information on six of the societies that were involved in the Root and Shoot project.
The ASPT and the BSA co-host their annual conferences, so the figures for attendance are the same.

| Name | Acronym | Total membership (2024) | Conference cadence | Attendance at last conference |
|---|---|---|---|---|
| American Society of Plant Biologists | ASPB | 2384 | Annual | 1417 |
| American Society of Plant Taxonomists | ASPT | NA | Annual | 852 |
| Botanical Society of America | BSA | 3126 | Annual | 852 |
| International Society for Molecular Plant-Microbe Interactions | IS-MPMI | 3801 | Every other year | 1133 |
| Maize Genetics Corporation | MGC | 477 | Annual | 441 |
| North American *Arabidopsis* Steering Committee | NAASC | 450 | Annual | 540 |

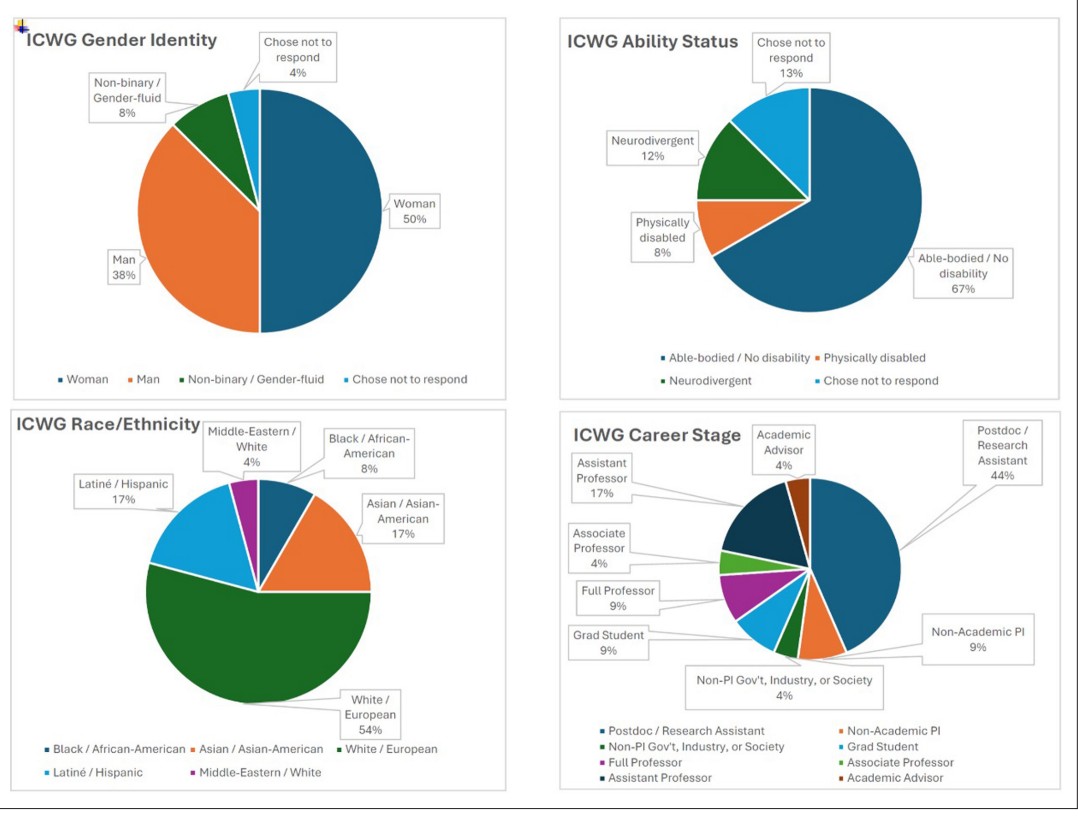

**Figure 1.** Inclusive Conference Working Group: demographic information. Membership of the Working Group broken down by gender identity (top left), ability status (top right), race/ethnicity (bottom left), and career stage (bottom right).

thousands of attendees. The final makeup of the Working Group is summarized in *Figure 1*.

We informed potential participants that the commitment would span one year, with a goal of completing the task in time for 2023 conferences, and that participants would need to dedicate six hours per month for the first six months. These hours were divided as follows: a one-hour monthly group discussion, with remaining hours spent on additional pre- and post-discussion reading, one to two subgroup meetings per month for specific tasks, and time for writing, editing, and communicating. In the second half of the year we expected the focus would shift to generating a final report for the plant science community and estimated the time commitment could be reduced to two to three hours per month (*Figure 2*). To compensate volunteers for their efforts, stipends were offered to participants. Some Working Group members chose to waive their stipends.

### *The process of guideline development*
The Inclusive Conference Working Group was divided into five subgroups. Each subgroup

independently developed a meeting style and approach to address one of the following:

1. A **community agreement** for conferences and participants outlining principles and standards that support equity and inclusiveness
2. A **reporting structure** that ensures accountability and compliance with the Community Agreement
3. Recommendations for a **transparent site selection** process that incorporates considerations regarding safety and inclusion of all potential attendees
4. Guidelines for practices that improve **conference accessibility**, including for those with disabilities and young families
5. Guidelines for **inclusive speaker selection and equitable programming**

## Recommendations for inclusive conferences
The final outcome of the process was a comprehensive set of recommendations to guide conference organizers in their planning. The recommendations

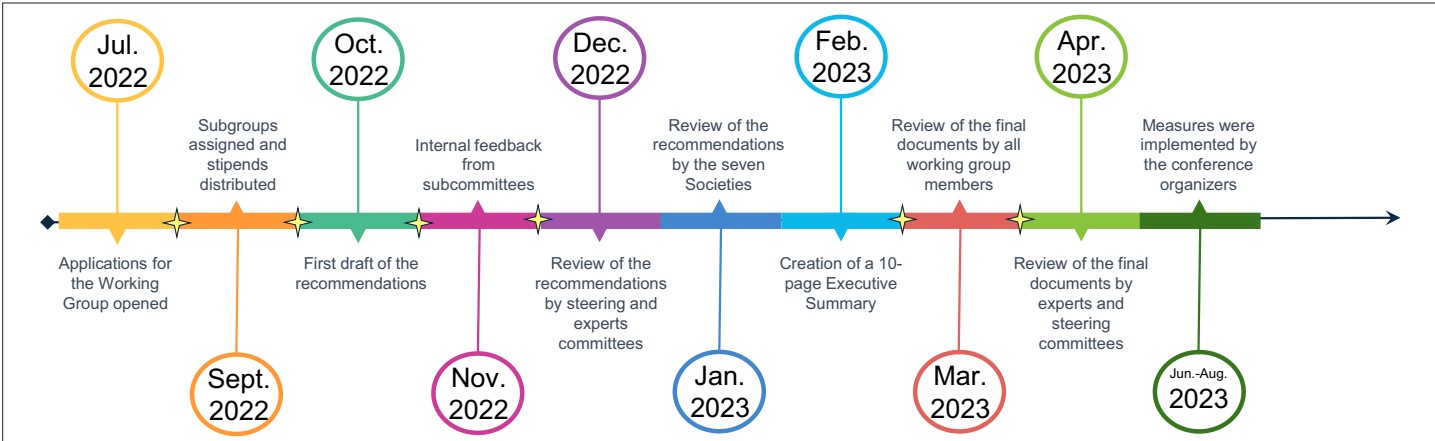

**Figure 2.** Timeline of the Working Group. The Inclusive Conferences Working Group ran from July 2022 until April 2023. The report containing the group's recommendations underwent three rounds of review: (i) internal review by the subgroups of the Working Group (November 2022); a strategic assessment by the ROOT & SHOOT steering committee, which contained at least one representative from each society (December 2022); (iii) a final evaluation by representatives from all seven participating societies, (January 2023).

include suggested language for community guidelines, guidance for reporting structures, advice for site selection and accessibility, and strategies for equitable programming and speaker selection. The guide, including the full Community Agreement, is available in both a long form (*Supplementary file 1*) and a short form (*Supplementary file 2*), and the key points are highlighted in *Table 2*.

Here we briefly summarize the key recommendations made by the Working Group, and reflect on any implementation or outcomes from our conferences.

### 1. Community agreement

Most conference attendees and society members have been asked to check a box indicating their agreement to abide by a Code of Conduct when registering for the conference. These Codes outline unacceptable behavior (e.g., harassment, intimidation, discriminatory or disruptive behavior) and some of the outcomes that such behaviors might lead to, which include verbal warning, expulsion from the event, and being barred from future events. The Working Group aimed to complement the existing punitive Codes of Conduct with a community agreement that states the shared values of the plant science community and emphasizes the *positive* behaviors expected of attendees (*Box 1*). Although there was a generally positive reaction to the tone of these recommendations, none of the conferences have adopted them as described, largely due to a perceived redundancy with current Codes of Conduct and a concern about whether the conference agreement could be enforceable. Nevertheless, at least one of the conferences is

currently planning to incorporate some of this language into their existing Code of Conduct.

### 2. Reporting structure

During community listening sessions, the need was repeatedly raised for a safe space where conference attendees could report disputes related to the Code of Conduct, or community agreement, and receive guidance. A central goal was to have a neutral party available for such reporting. The Working Group proposed that the ROOT & SHOOT grant provide on-site ombudsperson services to the society conferences, and the outcomes of this implementation are described further below. Bystander intervention training was also recommended and implemented as discussed below. Finally, the Working Group recommended that formal processes be established with clear guidelines for addressing reports of interpersonal conflicts and violence, not unlike the clear guidelines that are in place for academic misconduct and available from the Committee on Publication Ethics (COPE). At least one of the member societies has revised its processes to accommodate these recommendations.

The Working Group also recommended a public rating system whereby attendees could rate the accessibility of a conference after attending it: for example, they could score whether lactation rooms and closed captioning during oral presentations were provided. In an effort to measure the incidence of community agreement violations, the Working Group also recommended that a standardized survey be implemented for use both during and after the conference, for

**Table 2.** Summary of the major recommendations in the five areas addressed by the Inclusive Conference Working Group, based on a survey of the ASPB, BSA, IS-MPMI, MGC and NAASC.

Recommendations in place before 2023 are indicated by an open circle (one per conference), and those implemented in 2023 or later by a filled circle. See *Supplementary files 1 and 2* for more information.

| 1. Community agreement | 4. Conference accessibility | 5. Inclusive speaker selection and equitable programming |
|---|---|---|
| Create and share an aspirational community conference agreement OO● | Provide low-cost options for accommodation and registration OOOO | *Make a commitment* |
| Should be a collective vision of our values O● | Have a point person who reports on barriers and arranges accommodations ● | • Commit to inviting a diverse group of speakers; set measurable goals OOO● |
| Set expectations for conduct OOOO | Website, presentations, and signage should be accessible ●●● | • Commit to transparency: share these goals with speakers and participants O● |
| Define measures for accountability OOO | Ask participants about their accommodation needs in advance OOO● | *Form a diverse organizing committee* |
| | Be welcoming to parents and children OOO●● | • Emphasize commitment to diversity OOO● |
| **2. Reporting structure** | Provide lactation room, childcare, and family-friendly spaces OO● | • Recognize the organizing committee OO● |
| Provide bystander intervention training ●●●●● | Indicate allergens on food, and provide vegan and vegetarian options OOO● | *Assess equity goals* |
| Provide easy, confidential access to a reporting system OO●● | Provide quiet, fragrant-free spaces ●● | • Measure, track, and share progress O● |
| Provide an independent ombud onsite ●●●●● | Offer social events that don't include alcohol O● | • Survey the participants' experiences: do they feel respected, valued, and safe? O● |
| Have a clear system for dealing with transgressions OOO● | Check and consistently use correct speaker pronouns ● | *Broaden the scope of the meeting to provide space for all voices* |
| Conduct post-conference surveys OOOO | Use diacritical markers (like accents and tildes) appropriately O | • Don't have inclusion events run solely by people with historically excluded identities O●●● |
| | Ask about and use correct name pronunciation O● | • Treat speakers as scientists first OO●● |
| **3. Transparent site selection** | | *Create an equitable schedule* |
| Provide guidance and transparency for site selection O● | | • Be mindful of religious and cultural calendars, weekends OO●● |
| Costs: facilities rental, lodging, transportation OOOO | | • Plan for people to have breaks O●● |
| Accessibility: are the facilities ADA-compliant and able to provide additional accommodations if needed? OOOO● | | • Avoid scheduling conflicts, especially with those promoting a culture of inclusion O |
| Safety: availability of reproductive services ●● | | • When possible, include hybrid options O●● |
| Safety: are the streets safe to walk in after dark? OOO● | | *Provide necessary accommodations to speakers* |
| Safety: provide information about local LGBTQ+organizations O●●● | | • Physical access to stage, hearing loops, sign language interpreters ● |
| Consider convenience: are food options available nearby? OOO● | | • Gender neutral bathrooms at the venue O● |

reporting incidents and if/how it was addressed. These recommendations were not pursued for implementation by the professional societies, at least in part because organizers feared that such a rating system might preferentially penalize those societies with smaller conference budgets. Furthermore, the lack of clarity regarding details of the cross-society surveying (e.g., who would

gather and store data, how the data would be interpreted and communicated back to the societies) raised concerns, and thus, this aspect was not implemented.

### 3. Transparent site selection

The location of a conference affects participation and attendance in many ways. Many factors contribute to this decision, and the Working Group recommended being as transparent as possible in communicating how the site was selected. Ideally, the conference location is not unduly expensive so that registration costs are reasonable, although subsidies should be available to those who need them. Similarly, lower-cost accommodations should be available and promoted to the registrants. The venue must be accessible to individuals with visual, auditory, or mobility impairments. This information must be provided on the website, and accommodation requests must be honored. The conference should be held in a location that is safe for all (e.g., abortion and reproductive services are available, homosexuality is not illegal, transgender identities are respected) and where international travelers can easily obtain visas to visit. The Working Group developed a document with questions to guide site selection (*Supplementary file 3*), facilitating a process that ensures consideration of the identified issues and transparency for the community. Several conferences adopted some of these recommendations, particularly with regard to safety (see *Table 2*).

### 4. Conference accessibility

The Working Group considered five aspects of accessibility: economic, physical, audio/visual, family, and wellness. Collectively, these recommendations comprise the largest section of the Inclusive Conference Guide (*Supplementary file 4*), providing ample food for thought for conference organizers. These recommendations note that certain accessibility options (e.g., economic vs physical environment) may be in direct conflict, so a long-term view that balances choices over several years may be necessary. A key message is that conference organizers should strive to ensure that their event is open, accessible, and welcoming to all who wish to participate. Presenters should be provided with information on how to make their remarks accessible (*Supplementary file 4*), including closed captioning. All gender bathrooms must be available and clearly signposted. Accommodations should be made for people who attend with their children, such as affordable

onsite childcare and lactation rooms. With 20% or more of the population being neurodiverse, it is imperative to include dedicated quiet and scent-free spaces. Conferences provide ideal opportunities to educate the community about workspace accommodations. Finally, caterers should provide clearly labeled ingredient lists, including potential allergens, and ensure that inclusive dietary options are available to accommodate a wide range of needs and food restrictions.

### 5. Inclusive speaker selection and equitable programming

The final section provides concrete steps to ensure a diverse speaker pool across different career stages. We cannot claim that our conferences are inclusive if the presenters do not reflect the diversity of the audience. Conference organizers are encouraged to look beyond the senior authors of the most-cited recent papers if they want to achieve true speaker diversity. Inviting first authors, often early-career researchers and scholars, rather than senior authors, can be beneficial. Additionally, ensuring that the program includes topics that have not been historically represented – such as the role of Indigenous knowledge, science policy, the history of science, and education – can also be helpful. The program committee itself must be diverse, as studies show that diverse organizing committees are more likely to generate diverse speaker lists (*Segarra et al., 2020*). Being publicly and explicitly clear about a commitment to an inclusive conference increases the likelihood that invited speakers will accept an invitation. An inclusive approach also includes that those introducing speakers use correct pronouns (if provided) and make the effort to pronounce names accurately. Finally, being transparent about the commitment to equitable programming and speaker diversity can help with accountability, as can collecting year-by-year data and reflecting on trends as the organization prepares for the next conference.

## Implementing the recommendations

Although the Working Group developed a comprehensive set of recommendations, we recognize that not all have been or can be implemented, and some were already in place. Here, we highlight a few of the efforts that the ROOT & SHOOT project implemented across several conferences organized by the participating societies, along with their outcomes.

### Bystander intervention training workshops

From numerous listening sessions and discussions within the larger scientific community, personal safety at conferences emerged as one of the primary concerns of people with historically excluded identities. In order to begin to address this concern and build a more aware and supportive environment, ROOT & SHOOT partnered with the ADVANCEGeo Partnership to host several online bystander intervention training workshops. These workshops were free to attend and provided an early access point for individuals interested in understanding why, when, and how to intervene if observing situations involving mistreatment, harassment, or discrimination.

We have hosted seven training sessions since 2023, with an average attendance of 53 participants per session. As space was limited, organizations prioritized inviting those in leadership positions and session organizers to attend, after which the training was open to all, regardless of career level. Some organizations made attendance a requirement for their session chairs, while others strongly recommended it. These were extensive three-hour sessions, consisting of roughly two hours of training followed by an hour of practice, during which attendees actively practiced responding to problematic scenarios.

The practice portion was particularly well received, so we contracted ADVANCEGeo to hold additional 1.5 hour "scenario-only practice sessions" for those who had already undergone the full training. The length of the full training may have been a deterrent, so organizations that still wanted their members to be trained adapted the content into a shorter training session, and made it a requirement for conference organizers and session chairs to attend (*Friesner, 2023*). Organizations that did not make the training a requirement incentivized their members to join by providing badge stickers recognizing that they'd attended the training. While we did not conduct formal post-training evaluations, we received anecdotal feedback from members and conference participants who reported feeling more confident about intervening in problematic situations. Several also noted observing session chairs actively implementing the training practices during the conference.

### On-site ombudsperson and advocacy services

Another major finding from the Working Group was the need to have an independent, neutral party present at the conference to address personal safety needs and concerns about harassment and discrimination. Most affiliated societies were enthusiastic about engaging in such a service, provided that ROOT & SHOOT bore the cost. Therefore, at five conferences over the course of two years, ROOT & SHOOT paid for a professional ombudsperson and advocacy service.

The conference organizers informed attendees about the service through emails, conference applications, conference websites, and by introducing the ombudsperson/advocate during the opening activity. Attendees were provided with a web form and phone number to reach the on-site service provider, who had access to a private meeting space. When on-site, the services providers were proactive about being visible during the conference, ensuring that attendees were aware of the service and who the providers were.

The ombudsperson and advocacy service was requested and utilized at least once at each conference. On-site service providers could empathize with and guide the individuals who requested help. Equally valuable, they were able to inform conference leadership of issues that would likely have otherwise gone unnoticed while guiding leaders to help resolve issues via non-punitive, community-centered, restorative, and interest-based pathways.

To respect confidentiality, we can provide only general descriptions of the issues that arose. These issues included: helping process intrapersonal struggles that interfered with the individual's ability to be fully present and enjoy the conference; concerns about identity-based hate speech; sexual harassment; friction within a committee that involved unconscious bias related to identity; accessibility-related concerns; and racially biased material used in oral presentations. *Table 3* includes a series of situations that we have observed in our collective experience, along with potential ways to address them.

During the 2023 and 2024 conference seasons, the service providers met with conference leadership and staff via Zoom after each event to share their experience, answer questions, offer feedback to the conference organizers, and provide long-term recommendations for improving future conferences. These recommendations included: to review the presentation materials of major speakers; to have all speakers upload slides for review before the conference; to provide presentation templates that are accessible-friendly; and to have all attendees receive identity-based

**Table 3.** Summary of some challenges that have occurred during scientific conferences, and their proposed responses.

| Category | Example issue | Proposed response |
|---|---|---|
| Presentations | Offensive or stereotypical images, jokes, or language on slides | Speaker guidelines with content review |
| | Dismissive tone toward certain demographics during Q&A | Define a response beforehand and train the session chairs to intervene |
| Accessibility | Lack of captioning or sign language interpreters | Pre-event accessibility survey |
| | No accommodations for mobility or sensory needs | Provide multiple modes of participation |
| | Inaccessible venue layout | Visible signage and support at venue |
| Q&A sessions | Overly aggressive questioning, tone-policing, dismissive or personal attacks on speaker's competence or identity | Moderator training to set respectful tone |
| Networking events | Exclusion from informal groups, inappropriate jokes, harassment, or alcohol-related misconduct | Ombudsperson or advocacy ambassadors (with bystander training) at social events |
| Poster sessions | Harassment or inappropriate questioning during poster presentations | A clear system for reporting misconduct. Provide advocacy ambassadors with distinctive lanyards (with bystander training) at poster events |
| Social media and online platforms | Live posts or messages targeting individuals in harmful or sarcastic ways | Create and promote moderated hashtags |
| | Dogpiling or doxing (involves publicly exposing someone's private information) | Establish and promote a social media Code of Conduct |
| | Violation of Code of Conduct | Clear and visible Code of Conduct |
| General conduct | Public or private bullying | Advertise and make the ombudsperson role visible. Provide and advertise bystander intervention training |
| | Microaggressions | Consistent follow-up for violations, consider implementing restorative justice methods |
| Power dynamics | Abuse of power by established figures (inappropriate comments, gatekeeping, retaliation against dissent) | Policy outlining professional boundaries and recourse for abuse. Provide and advertise bystander intervention training |

bias training, as well as bystander intervention training.

While we have limited data on post-conference evaluations from two organizations, most of the feedback we received was informal, and was shared with organizers and the service providers throughout the conference. In one conference, at least five people stopped by the office being used by the service provides to say that they did not need the service, but were glad it was available. The same conference's evaluations showed that 66% of members were aware that the service was available to them (strongly agree +somewhat agree; n=167), and 63% understood the purpose of the service (strongly agree +somewhat agree; n=170). Some 47% said that having the service was important to them, while 39% were neutral and 13% disagreed or strongly disagreed (n=165). In another conference, evaluations showed that 35% considered the service to be of "high value" (between 8 and 10 on a scale of 1–10; n=191).

We acknowledge that cost is a significant barrier to having an onsite ombudsperson and advocacy service at (typically between $8,000 and $15,000 per conference). While ROOT & SHOOT had the benefit of NSF funding, organizations that may want to employ a similar service might not have the funds to do so. Although they may be more limited in the current political climate, organizations can consider seeking funding from private foundations that focus on diversity, equity, inclusion and accessibility, or teaming up with other conference organizers and trying to negotiate a discount when booking services for a number of conferences.

Another option is to book a remote service, where the ombudsperson/advocate attends the conference virtually and is available via phone, email, or text. A virtual service typically costs between $3,000 and $ 5,000 per conference.

### Exploring more just systems to respond to harm

The Working Group was also made aware of concerns regarding the systems in place for addressing reports and instances of harm at conferences. We are aware of commonly reported

## Box 1. Proposed text for a community agreement.

The following templates for a statement of community values and a conference participant agreement were developed by the Inclusive Conferences Working Group.

### Community values

[CONFERENCE NAME] strives to ensure the full participation, safety, and support of all participants. We work to uphold inclusive and welcoming antiracist and decolonizing practices and values; to center and support queer, trans, non-binary, and gender minority scientists; to provide equitable access for scientists with disabilities; to invite and uplift contributions by scientists of all religions, belief systems, and cultural backgrounds; to elevate the work of researchers in underserved communities and universities; to strive for more equitable access to resources in those communities; and to open doors for those whom science has historically excluded.

### Conference participant agreement

As a participant in [CONFERENCE NAME], I agree to do my part in building a welcoming and inclusive conference environment by:

- Upholding and holding others accountable to the above-described community values while attending formal events as well as unofficial activities while at the conference.
- Critically evaluating implicit biases and reflecting on how they impact our interactions.
- Treating others with civility and actively working to avoid, prevent, stop harassment, discrimination, and behavior that is threatening, intolerant, abusive or violent.
- Upholding the highest standards of scientific and academic integrity during the conference, recognizing past and present contributors to science, while acknowledging that science can be flawed and historically has harmed marginalized communities.
- Evaluating the work of colleagues equitably—with open-minded fairness and respect.
- Encouraging and promoting climates where multiple scientific perspectives may be freely expressed and valued.

---

outcomes, including no action being taken following a complaint, a lack of transparency regarding the process, and no follow-up with the complainant (see, for example, *Ahmed, 2021*). We asked, in an inclusive, transparent, empowered, and equal plant science culture, what would happen when a transgression occurred? Answering that question requires involving the community in establishing guidelines and consequences. Ideally, we would like to see processes that are not purely punitive, but rather those through which relationships are repaired and that allow those who have experienced harm to reclaim their agency. Therefore, we explored the possibility of incorporating restorative justice models. Restorative justice is an approach to addressing harm or the risk of harm by engaging all those affected in reaching a common understanding and agreement on how the harm or wrongdoing can be repaired and justice achieved

(*Zehr, 2002*). Restorative justice focuses not on prosecution and punishment but on harm done and how to repair it. It expands the circle of stakeholders beyond the governance and offender to include victims and community members. Importantly, restorative justice has been shown to help the transgressors recognize the impact of their actions and work to make meaningful amends to those they have harmed.

In early 2024, we initiated conversations with the plant science community about how we collectively respond to incidents of harm, not only to hear their ideas but also to build trust and goodwill, so that people would be more likely to comply with the processes that were developed. Following several community listening sessions and conversations facilitated by two of the authors (LM and BM), we have drafted an initial accountability process guide and decision-making tree, which we are continuing to refine and develop.

However, we note that although a small number of individuals have been very engaged in these conversations, the majority of our community members have not participated in this process; we acknowledge that our community may not be ready to fully engage in this discussion.

## Lessons learned and future challenges

By engaging stakeholders from diverse backgrounds, different societies, and career stages, we increased members' sense of belonging within these organizations, especially amongst early-career scientists. Engaging diverse stakeholders and considering their own lived experiences raised questions and issues that had not traditionally been included in the conference planning process. For instance, concerns such as dietary restrictions, personal safety due to conference location, and scheduling conflicts during religious and cultural events were highlighted by the Working Group. Not only were the recommendations developed by the Working Group helpful for organizers in planning conferences, but the process of community engagement to create a shared, iterative set of recommendations also increased the commitment by the organizations to implement them further.

One of the main challenges of this Working Group was coordinating a team of 30 people across multiple time zones, which made it difficult to schedule meetings where everyone could participate. In the future, a smaller core group with clearly defined roles could improve efficiency while still allowing for broad input through periodic check-ins with a larger advisory network. Additionally, the subgroup structure was intentionally flexible, allowing members to self-organize based on interest and expertise. While this approach encouraged engagement, a more structured framework with clearer expectations and timelines might have helped streamline the process. Another key lesson was the overall timeline – completing the work took longer than anticipated. A more iterative approach, gathering feedback from multiple rounds of conferences, could provide better insights into what can realistically be implemented and how effective different recommendations are in practice.

Most of the challenges from this work have arisen from individual societies agreeing with the recommendations in principle but experiencing various barriers to implementing them in practice at the conferences they organize, as outlined below.

Several societies raised concerns about monetary costs. The ROOT & SHOOT project includes large and small organizations with different administrative support systems and funding models. Although some Working Group recommendations can be implemented through policy changes alone, others – such as providing onsite American Sign Language (ASL) interpreters or ombudsperson/advocate services – incur significant budgetary costs. As an example, the ASL interpreter required scientific training to accurately convey technical content, a level of specialization that is not mandated by law but was necessary for meaningful access. This increased both the interpreter's fee and travel-related costs. Additional logistical accommodations, such as ensuring clear visual lines and coordinated session planning between the interpreter and the speaker, were also required. Additionally, legal requirements around accessibility vary by country, and international conferences may not be covered under the same mandates as domestic events. By funding some of these efforts through the ROOT & SHOOT project, we aimed to demonstrate their value and encourage individual societies to sustain their implementation beyond the funding period. Budgetary concerns are likely to persist, and by making the case to funding agencies that supporting the specific items was shown to be effective, this will better enable societies to maintain and expand the inclusivity of their conferences.

One society decided not to have an on-site ombudsperson as they felt that doing so would duplicate existing efforts; this society also had liability concerns associated with contracting a group outside of their society. On the other hand, two conferences that benefited from the on-site provider have elected to find funds to provide this service in future years.

Another challenge we encountered was assessing the outcomes of this work, as different societies employ very different conference assessment tools. Although we requested specific questions to be included in these tools, these requests were generally unsuccessful. In cases where we were able to gather data, we found that small numbers of individuals engaged with the bystander training or onsite provider, which raises the question of how to interpret and proceed with such data. When budget decisions are made, how should an intervention that has a considerable positive impact on a small number of people be ranked?

## Conclusions

This project was undertaken from 2022–2024,, during which the NSF actively supported programs to foster inclusion, promote equity, and enhance accessibility. Although the short-term prospects for continuing such work with federal funding in the US looks bleak, we emphasize that most of the recommendations – such as providing an aspirational community agreement that focuses on positive values, providing attendees with information about accessibility, labeling food ingredients, providing family-friendly spaces and gender-neutral bathrooms, and creating a diverse and equitable program – can be implemented without significant cost to the organizers or participants. It is incumbent on all community-serving organizations to work inclusively, to the best of their abilities, so that all community members feel welcome and safe. Community volunteers can accomplish many of these tasks by pre-screening slides for accessibility, surveying the meeting site to provide accessibility information, and serving as onsite advocates. Efforts to promote inclusion at conferences can be carried out discreetly if necessary. Our experiences show us that those who benefit will be aware of the efforts made - those who know, know.

**Marcia Puig-Lluch** is at the ROOT & SHOOT Research Coordination Network, Rockville, United States
https://orcid.org/0009-0009-4063-2364

**Mary Williams** is at the American Society of Plant Biologists, Rockville, United States
https://orcid.org/0000-0003-4447-7815

**Eric Wada** is at Folsom College, Folsom, United States, and is a member of the Botanical Society of America
https://orcid.org/0009-0003-6928-0776

**Imeña Valdes** is at Northwestern University, Evanston, and the Chicago Botanic Garden, Glencoe, United States, and is a member of the Botanical Society of America
https://orcid.org/0009-0003-4209-0331

**Carrie Tribble** is at the University of Washington, Seattle, United States, and is a member of the American Society of Plant Taxonomists and the Botanical Society of America

**Andrew Read** is at the University of Minnesota, Saint Paul, United States, and is a member of the International Society for Molecular Plant-Microbe Interactions

**Kanwardeep S Rawale** is at Geneshifters, Pullman, United States, and is a member of the American Society of Plant Biologists
https://orcid.org/0000-0003-3894-4712

**Chelsea L Newbold** is at Pennsylvania State University, University Park, United States, and is a member of the American Phytopathological Society
https://orcid.org/0000-0002-2765-1503

**Bathabile Mthombeni** is at Untangled Resolutions, Binghampton, United States

**Michael Moody** is at the Botanical Society of America, St. Louis, United States
https://orcid.org/0000-0003-0327-267X

**Laura Minero** is at Alchemy Psychology Colectivo Inc, Los Angeles, United States

**Annarita Marrano** is at AgBioData Consortium & Phoenix Bioinformatics, Fremont, and the ROOT & SHOOT Research Coordination Network, Rockville, United States
https://orcid.org/0000-0001-9560-2706

**Melanie Link-Perez** is at the Wisconsin State Herbarium, University of Wisconsin–Madison, Madison, United States, and is a member of the Botanical Society of America
https://orcid.org/0000-0002-5435-5750

**Roger W Innes** is at Indiana University, Bloomington, United States, and is a member of the International Society for Molecular Plant-Microbe Interactions
https://orcid.org/0000-0001-9634-1413

**Cody Coyotee Howard** is at Oklahoma State University, Stillwater, United States, and is a member of Botanical Society of America
https://orcid.org/0000-0001-7662-9102

**Adriana Hernandez** is at Cornell University, Ithaca, United States, and is a member of Botanical Society of America

**Corri Hamilton** is at the University of Missouri, Columbia, United States, and is a member of the American Phytopathological Society

**Denita Hadziabdic** is at the University of Tennessee, Knoxville, United States, and is a member of the American Phytopathological Society
https://orcid.org/0000-0003-1991-2563

**Morgan Gostel** is at the Botanical Research Institute of Texas, Fort Worth, United States, and is a member of the American Society of Plant Taxonomists and the Botanical Society of America
https://orcid.org/0000-0002-3169-627X

**Joanna Friesner** is at the North American Arabidopsis Steering Committee, Corvallis, United States
https://orcid.org/0000-0002-6799-7808

**John E Fowler** is at the Oregon State University, Corvallis, United States, and is a member of Maize Genetics Cooperation
https://orcid.org/0000-0002-9254-5934

**Mindy Findlater** is at the University of California, Merced, Merced, United States, and is a member of the American Society of Plant Biologists

**Sakina Elshibli** is at the University of Helsinki, Helsinki, Finland, and is a member of the American Society of Plant Biologists
https://orcid.org/0000-0002-3292-1029

**Steven J Burgess** is at the University of Illinois at Urbana–Champaign, Urbana, United States, and is a member of the American Society of Plant Biologists
https://orcid.org/0000-0003-2353-7794
**Hank W Bass** is a plant biologist at a research university in the United States. He is writing in a personal capacity and not on behalf of his employer or institution
**Burcu Alptekin** is at the University of Wisconsin–Madison, Madison, United States, and is a member of the American Society of Plant Biologists
**R Shawn Abrahams** is at the University of Illinois at Urbana–Champaign, Urbana, United States, and is a member of the Botanical Society of America
**Patricia Baldrich** is at the University of California, Davis, United States and is a member of the International Society for Molecular Plant-Microbe Interactions
pbaldrich@ucdavis.edu
https://orcid.org/0000-0003-4669-6632

*Author contributions:* Marcia Puig-Lluch, Writing – original draft, Participated in the working group; Mary Williams, Writing – original draft, Participated in the working group; Eric Wada, Participated in the working group; Imeña Valdes, Writing – original draft, Participated in the working group; Carrie Tribble, Participated in the working group; Andrew Read, Writing – original draft, Participated in the working group; Kanwardeep S Rawale, Participated in the working group; Chelsea L Newbold, Participated in the working group; Bathabile Mthombeni, Writing – original draft, Participated in the working group; Michael Moody, Participated in the working group; Laura Minero, Writing – review and editing, Participated in the working group; Annarita Marrano, Participated in the working group; Melanie Link-Perez, Participated in the working group; Roger W Innes, Writing – original draft, Participated in the working group; Cody Coyotee Howard, Participated in the working group; Adriana Hernandez, Participated in the working group; Corri Hamilton, Participated in the working group; Denita Hadziabdic, Writing – original draft, Participated in the working group; Morgan Gostel, Writing – original draft, Participated in the working group; Joanna Friesner, Participated in the working group; John E Fowler, Writing – original draft, Participated in the working group; Mindy Findlater, Writing – original draft, Participated in the working group; Sakina Elshibli, Participated in the working group; Steven J Burgess, Participated in the working group; Hank W Bass, Participated in the working group; Burcu Alptekin, Writing – original draft, Participated in the working group; R Shawn Abrahams, Participated in the working group; Patricia Baldrich, Conceptualization, Writing – original draft, Writing – review and editing, Participated in the working group

*Competing interests:* The authors declare that no competing interests exist.

## Funding

| Funder | Grant reference number | Author |
|---|---|---|
| National Science Foundation | DBI-2134321 | Mary Williams |

The funders had no role in study design, data collection and interpretation, or the decision to submit the work for publication.

**Decision letter and Author response**
Decision letter https://doi.org/10.7554/eLife.106877.sa1
Author response https://doi.org/10.7554/eLife.106877.sa2

## Additional files

**Supplementary files**
Supplementary file 1. Full set of Working Group recommendations.

Supplementary file 2. Executive summary of Working Group recommendations.

Supplementary file 3. Check list for site selection.

Supplementary file 4. Conference accessibility guide.

## Data availability

There are no data associated with this work.

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
