## [Decision Letter]

**Decision letter after peer review:**

Thank you for submitting your article " Engaging Stakeholders in Conference Design to Promote Inclusion in the Plant Sciences" to *eLife* for consideration as a Feature Article.

Your article has been reviewed by two peer reviewers, and the evaluation has been overseen by Peter Rodgers of *eLife*. The following individual involved in review of your submission have agreed to reveal their identity: Yvette Seger (Reviewer #2). Please note that Reviewer #1 did not submit a report.

The reviewers and editor have discussed the reviews and we have drafted this decision letter to help you prepare a revised submission. The length of the article is not an issue, so please take as many words as you need (and include Figures as appropriate) to address the comments from the reviewers.

Summary

This article documents the outcomes of an NSF-funded initiative called ROOT & SHOOT that was intended to foster diversity, equity, accessibility and inclusion in scientific conferences organized by scientific societies in the field of plant biology. The article will be a useful resource for anyone seeking to organize a conference that will be inclusive and welcoming for individuals of all backgrounds and career stages. However, there are a number of points that need to be addressed, as outlined below.

General assessments

*Reviewer #2:*

This manuscript documents the outcomes of an NSF-funded initiative intended to foster diversity, equity, accessibility and inclusion (DEAI) of the conferences of a collective of seven societies in plant biology. While initiated by the plant biology community, the challenges associated with making scientific conferences inclusive and welcoming environments for individuals of all backgrounds and career stages are not unique to plant biology disciplines, and thus the working group approach and resulting strategies are broadly applicable across STEM fields. This report-out provides an important overview of a consensus-based strategy to obtain member input that was used to make the conferences of the participating societies safe and inclusive environments for fostering professional networks and communicating scientific findings.

*Reviewer #3:*

The authors and collaborators undertook an ambitious effort to gather feedback and construct recommendations for making scholarly conferences more inclusive. The manuscript conveys some important information regarding both the process and the outcomes of their work. I want to express my admiration for the ROOT & SHOOT team and all they have accomplished. My comments below are in the spirit of collaboration, with the hope of reshaping the manuscript (slightly!) so their work reaches as broad an audience as possible.

Essential revisions:

[1] As it is a review of a process utilized by the ROOT & SHOOT collaborative, there are no obvious weaknesses. However, recognizing that this essay could be a resource used by other scientific organizations to address comparable concerns at their conferences, it would be useful to include additional information about the participating societies, such as approximate total membership for each, conference cadence, and approximate conference attendees, allowing readers to better understand the scope of this effort.

[2] The authors also acknowledge that several of the agreed upon strategies were only feasible due to the ability to defray associated costs via the NSF grant. The conclusion alludes to current challenges associated with obtaining federal support for these activities. In addition to highlighting the low-cost options that organizations can pursue to make a conference more inclusive, it would be helpful to share strategies considered by the ROOT & SHOOT consortium to defray costs associated with bystander training and/or on-site ombud/advocate to continue these interventions following the conclusion of grant support.

[3] The biggest weakness of the manuscript is that it felt like it was telling two stories (1 process and 2 outcome), and in my opinion, focused too heavily on the process. As such, the article would benefit from a longer discussion of the outcomes and recommendations.

The recommendations were largely relegated to a table, with little elaboration. The authors elaborated on only three interventions – but did not explain why those three? Were these the "most likely to be successful for the largest number of conferences"? Were they the most widely trialled by ROOT&SHOOT societies?

I was also intrigued by the note that some of the ICWG recommendations were already being implemented by societies. That's great! It would be amazing to list those (or simply add an asterisk in the table indicating they are already in-place by societies (maybe if 2 or more are already doing it?)).

At the end of the manuscript, you mention some of the interventions are low-cost – can you somehow point those out in the table as well?

My biggest wish for this manuscript would be for the group to make it really obvious what the low-hanging fruits are that all conferences could implement.

[4] My burning question when reading about the ICWG was if anyone on the WG had ever organized a conference before (perhaps serving on a conference organizing committee) or if any professional conference staff from participating societies engaged with the WG at all (as members or guests). My concern about this grew when the manuscript included legally-required ADA compliance (ASL interpreters) in a sentence about how costs may prevent future implementation of some recommendations. Any professional event staff know and understand that this is a legal requirement (with very few exceptions). Of the things I've listed to be added, this would be my #1 request.

[5] "On-site ombudsperson and advocacy ambassador" section: the paragraph starting with "due to confidentiality standards of practice…" has information that would be great to highlight more. We talk about making societies more inclusive and speak in vague terms about how conferences are "not inclusive", but majority attendees do not understand what that looks like. A text box highlighting these would be impactful in showing exactly what issues arise at real conferences.

[6] "On site ombudsperson" section: A minor recommendation that could improve this or future presentations around the results would be to elaborate on usage of what sounded like a very costly service. In the manuscript, you note that this service was requested and utilized "at least once" at each conference. Given the expense involved, it would be helpful to have some sort of metric – e.g., number of inquiries per thousand attendees per conference day – or something like that. I realize that is watering down a lot of important information and assigning metrics to something like providing a safe and inclusive environment is maybe questionable, but the reality is that conferences are getting more and more expensive to provide what we already provide (AV, catering, security, etc. have all gone through the roof in recent years).

Similarly, you mention a society opted out since they already had a procedure in place – can ROOT & SHOOT facilitate some sort of comparison of costs, effectiveness, utilization, etc. for the different models of having some reporting/ombudsperson function at a conference? What sort of procedures and training could the society that does this on their own share with others who can't afford professional services in the future?

[7] Bystander intervention training section: What was attendance like in these sessions? Were society leaders required to attend or was attendance "preaching to the choir"? (I do not mean that in a harsh way; as someone who has worked in this space, I mean that voluntary attendance rarely attracts the people who need to be there).

[8] "Working towards a more just system of accountability". I have done a lot of work in the DEI space and I still struggled to understand in practical terms what this section was trying to convey. Even using a hypothetical example would be helpful here.

[9] I was very interested in reading more about the implementation phase and seeing more detail about how the recommendations translated into new practices. What did attendees say? Did they find the bystander intervention training valuable? How many people signed up? Did conferences with high bystander intervention training participation see more use of reporting of incidents? Did attendees at those conferences report a higher sense of belonging at the conference?

I am completely sympathetic to the authors' comment that trying to get standardized language into conferences run by multiple groups was challenging, but even seeing post-event data from 1-2 conferences would have been very powerful – even some qualitative open ended responses indicating attendee reaction to those interventions (e.g., x percent/number of attendees specifically mentioned inclusive features in open-ended survey questions following conferences).

---

## [Author Response]

Essential revisions:[1] As it is a review of a process utilized by the ROOT & SHOOT collaborative, there are no obvious weaknesses. However, recognizing that this essay could be a resource used by other scientific organizations to address comparable concerns at their conferences, it would be useful to include additional information about the participating societies, such as approximate total membership for each, conference cadence, and approximate conference attendees, allowing readers to better understand the scope of this effort.

We appreciate the suggestion to provide additional context about the participating societies. To address this, we have created a new supplemental table (Supplemental Table 1) that includes key logistical details for each society, such as total membership, frequency of conferences, and typical conference attendance. We invited all participating societies to contribute this information via a shared document, which we compiled into the table now included in the revised manuscript. This added context helps clarify the scale and diversity of the ROOT & SHOOT collaborative effort.

[2] The authors also acknowledge that several of the agreed upon strategies were only feasible due to the ability to defray associated costs via the NSF grant. The conclusion alludes to current challenges associated with obtaining federal support for these activities. In addition to highlighting the low-cost options that organizations can pursue to make a conference more inclusive, it would be helpful to share strategies considered by the ROOT & SHOOT consortium to defray costs associated with bystander training and/or on-site ombud/advocate to continue these interventions following the conclusion of grant support.

We agree that sustainability of these interventions beyond the period of federal support is an important concern. To address this, we have expanded the “On-site ombudsperson and advocacy ambassador” section to include a discussion of potential cost-sharing strategies among participating societies, including the idea of a joint contract model to distribute costs across multiple annual meetings. We also note that private foundations with a focus on DEIA and accessibility may be well-positioned to support these types of services going forward.

In addition, we suggest lower-cost alternatives such as engaging a virtual ombudsperson or advocate who can attend sessions remotely and remain accessible via phone or text. This approach could significantly reduce expenses related to travel and lodging while maintaining a critical layer of support for attendees. These options, along with detailed cost breakdowns and implementation considerations, are now included in the revised text to help other organizations plan for long-term integration of these essential services.

[3] The biggest weakness of the manuscript is that it felt like it was telling two stories (1 process and 2 outcome), and in my opinion, focused too heavily on the process. As such, the article would benefit from a longer discussion of the outcomes and recommendations.The recommendations were largely relegated to a table, with little elaboration. The authors elaborated on only three interventions – but did not explain why those three? Were these the "most likely to be successful for the largest number of conferences"? Were they the most widely trailed by ROOT&SHOOT societies?I was also intrigued by the note that some of the ICWG recommendations were already being implemented by societies. That's great! It would be amazing to list those (or simply add an asterisk in the table indicating they are already in-place by societies (maybe if 2 or more are already doing it?)).At the end of the manuscript, you mention some of the interventions are low-cost – can you somehow point those out in the table as well?My biggest wish for this manuscript would be for the group to make it really obvious what the low-hanging fruits are that all conferences could implement.

Thank you for this thoughtful and constructive feedback. In response, we have expanded the discussion of outcomes and recommendations throughout the manuscript. Specifically, the section “INCLUSIVE CONFERENCE RECOMMENDATIONS” now includes five distinct subsections corresponding to the major outcome categories and the subgroups that contributed to them. This provides a more comprehensive narrative of the process-to-outcome pipeline. We also revised Table 1 to indicate which recommendations were already being implemented prior to the ICWG effort, as well as which were adopted afterward. In addition, we included new supplemental materials, such as a checklist for Transparent site selection and the recommendations for Accessibility. These additions aim to make the manuscript a more practical and immediately useful resource for other scientific organizations.

[4] My burning question when reading about the ICWG was if anyone on the WG had ever organized a conference before (perhaps serving on a conference organizing committee) or if any professional conference staff from participating societies engaged with the WG at all (as members or guests). My concern about this grew when the manuscript included legally-required ADA compliance (ASL interpreters) in a sentence about how costs may prevent future implementation of some recommendations. Any professional event staff know and understand that this is a legal requirement (with very few exceptions). Of the things I've listed to be added, this would be my #1 request.

We have added the information of the general experience of the working group participants. “Although it was not a requirement, most (seventeen out of twenty-four) of those who participated in the Working Group had previous experience organizing a conference. Their experiences ranged from having organized informal networking events and smaller workshops to having professionally organized several in-person and virtual conferences with thousands of attendees.”

We also added some of the ASL challenges that we observed based on our experience in the challenge section “As an example, the ASL interpreter required scientific training to accurately convey technical content, a level of specialization that is not mandated by law but was necessary for meaningful access. This increased both the interpreter's fee and travel-related costs. Additional logistical accommodations, such as ensuring clear visual lines and coordinated session planning between interpreter and speaker, were also required. Furthermore, legal requirements around accessibility vary by country, and international conferences may not be covered under the same mandates as domestic events.”

[5] "On-site ombudsperson and advocacy ambassador" section: the paragraph starting with "due to confidentiality standards of practice…" has information that would be great to highlight more. We talk about making societies more inclusive and speak in vague terms about how conferences are "not inclusive", but majority attendees do not understand what that looks like. A text box highlighting these would be impactful in showing exactly what issues arise at real conferences.

Thank you for this important suggestion. We have created a new table (Table 2) that outlines a range of real or commonly observed incidents that can compromise inclusivity at conferences. These include inappropriate or offensive slide content, lack of accessibility accommodations, disrespectful or aggressive behavior during Q&A sessions, violations of the code of conduct, and exclusionary dynamics during networking events. For each issue, we provide a brief description along with potential strategies for prevention or response. These examples draw from both our collective experience and scenarios presented in bystander training sessions. Our goal is to offer readers a clearer picture of what non-inclusive environments can look like in practice, and how societies and organizers can proactively address them. We believe that by making these situations visible and actionable, scientific organizations will be better equipped to foster safer, more welcoming conference environments.

[6] "On site ombudsperson" section: A minor recommendation that could improve this or future presentations around the results would be to elaborate on usage of what sounded like a very costly service. In the manuscript, you note that this service was requested and utilized "at least once" at each conference. Given the expense involved, it would be helpful to have some sort of metric – e.g., number of inquiries per thousand attendees per conference day – or something like that. I realize that is watering down a lot of important information and assigning metrics to something like providing a safe and inclusive environment is maybe questionable, but the reality is that conferences are getting more and more expensive to provide what we already provide (AV, catering, security, etc. have all gone through the roof in recent years).Similarly, you mention a society opted out since they already had a procedure in place – can ROOT & SHOOT facilitate some sort of comparison of costs, effectiveness, utilization, etc. for the different models of having some reporting/ombudsperson function at a conference? What sort of procedures and training could the society that does this on their own share with others who can't afford professional services in the future?

We appreciate this thoughtful suggestion. To address it, we have added a paragraph at the end of the “On-site ombudsperson and advocacy ambassador” section that outlines typical cost structures for this service. In-person ombudsperson services generally range from $8,000–$15,000 per conference, depending on the level of engagement. This estimate includes daily professional fees, registration, travel, lodging, and per diem. In contrast, virtual participation models (including phone or text-based support) typically range from $3,000–$5,000 and represent a more accessible option for smaller meetings or societies with limited budgets.

[7] Bystander intervention training section: What was attendance like in these sessions? Were society leaders required to attend or was attendance "preaching to the choir"? (I do not mean that in a harsh way; as someone who has worked in this space, I mean that voluntary attendance rarely attracts the people who need to be there).

Intervention Training Workshops” that discusses attendance and participation in more detail. Attendance was voluntary and varied by society and conference, and we acknowledge that this approach may not always reach individuals who would benefit most from the training. Some societies marked attendees with a different color badge, which created visibility but also raised concerns about signaling. To address this issue going forward, societies are exploring strategies to encourage broader participation, including offering incentives such as discounted registration for attendees who complete training. In one case, a society required bystander intervention training for individuals organizing sessions, a model we highlight as a promising step toward structural change.

We have also included a new supplemental resource, a slide deck developed by one of our working group members that can serve as a starting point for societies interested in implementing their own training sessions. Our goal is to promote both accessibility and accountability in the adoption of these practices.

[8] "Working towards a more just system of accountability". I have done a lot of work in the DEI space and I still struggled to understand in practical terms what this section was trying to convey. Even using a hypothetical example would be helpful here.

Thank you for this valuable feedback. We agree that this section addresses a complex and evolving topic, and we appreciate the opportunity to clarify our intent. In response, we have added a section named “Exploring More Just Systems to Respond to Harm.” This section includes a clearer explanation of restorative justice principles, emphasizing the importance of centering the needs of the person harmed while also holding the person who caused harm accountable in a way that prevents future harm. We acknowledge that our community is still in the early stages of exploring these frameworks. As part of this process, we held listening sessions to better understand readiness and perspectives across our member societies. These sessions revealed both enthusiasm for change and significant uncertainty about implementation, which we reflect in the revised text.

[9] I was very interested in reading more about the implementation phase and seeing more detail about how the recommendations translated into new practices. What did attendees say? Did they find the bystander intervention training valuable? How many people signed up? Did conferences with high bystander intervention training participation see more use of reporting of incidents? Did attendees at those conferences report a higher sense of belonging at the conference?I am completely sympathetic to the authors' comment that trying to get standardized language into conferences run by multiple groups was challenging, but even seeing post-event data from 1-2 conferences would have been very powerful – even some qualitative open ended responses indicating attendee reaction to those interventions (e.g., x percent/number of attendees specifically mentioned inclusive features in open-ended survey questions following conferences).

Thank you for this thoughtful and important comment. We have expanded the “On-site Ombudsperson and Advocacy Ambassador” section to include additional details on how these interventions were received by attendees. While it is still early in the implementation phase and comprehensive quantitative data across conferences are not yet available, we do have preliminary feedback from post-event surveys and informal debriefs.